ecology

above–belowground interactions, desert, dryland decomposition conundrum, litter decomposition, macro-detritivores, terrestrial isopod

**Author for correspondence:**
Nevo Sagi
e-mail: nevo.sagi@mail.huji.ac.il

# Burrowing detritivores regulate nutrient cycling in a desert ecosystem

Nevo Sagi[1], José M. Grünzweig[2] and Dror Hawlena[1]

[1]Risk-Management Ecology Lab, Department of Ecology, Evolution and Behavior, The Alexander Silberman Institute of Life Sciences, The Hebrew University of Jerusalem, Jerusalem 91905, Israel
[2]Robert H. Smith Institute of Plant Sciences and Genetics in Agriculture, Robert H. Smith Faculty of Agriculture, Food and Environment, The Hebrew University of Jerusalem, Rehovot, Israel

NS, 0000-0002-4745-4304

Nutrient cycling in most terrestrial ecosystems is controlled by moisture-dependent decomposer activity. In arid ecosystems, plant litter cycling exceeds rates predicted based on precipitation amounts, suggesting that additional factors are involved. Attempts to reveal these factors have focused on abiotic degradation, soil–litter mixing and alternative moisture sources. Our aim was to explore an additional hypothesis that macro-detritivores control litter cycling in deserts. We quantified the role different organisms play in clearing plant detritus from the desert surface, using litter baskets with different mesh sizes that allow selective entry of micro-, meso- or macrofauna. We also measured soil nutrient concentrations in increasing distances from the burrows of a highly abundant macro-detritivore, the desert isopod *Hemilepistus reaumuri*. Macro-detritivores controlled the clearing of plant litter in our field site. The highest rates of litter removal were measured during the hot and dry summer when isopod activity peaks and microbial activity is minimal. We also found substantial enrichment of inorganic nitrogen and phosphorous near isopod burrows. We conclude that burrowing macro-detritivores are important regulators of litter cycling in this arid ecosystem, providing a plausible general mechanism that explains the unexpectedly high rates of plant litter cycling in deserts.

## 1. Introduction

A key question in arid land ecology is why plant litter does not accumulate over time [1]. Deserts are characterized by little and highly variable precipitation amounts, and extreme surface temperatures [2]. These conditions are unfavourable for decomposer activity [3], and therefore global decomposition models predict very low rates of plant litter decomposition under arid conditions [4]. However, empirical studies using microbial litter bags found that the observed decomposition rates exceed the rates predicted by these models [5,6]. This discrepancy was termed the dryland decomposition conundrum [7].

Attempts to reconcile this conundrum have predominantly focused on abiotic weathering agents, such as photodegradation [8] and thermal degradation [9], and also on alternative sources of moisture such as fog [10], dew and atmospheric water vapour [11]. Soil–litter mixing and precipitation pulse frequency were also suggested as mechanisms that regulate decomposition in drylands [12]. These attempts yielded convincing evidence that such processes may indeed increase rates of aboveground litter decomposition, reducing the discrepancy between the models and the litter-bag experiments [13,14]. However, the importance of these mechanisms in explaining the overall rates of plant litter cycling outside the litter-bag realm remains largely unknown.

Noy-Meir [1] suggested another explanation for the high rate of litter cycling in deserts, emphasizing the important role of 'macrodecomposition' by detritivorous arthropods. Macro-detritivores are known to assist plant litter cycling in many terrestrial ecosystems [15]. They do so by consuming and assimilating plant litter

nutrients and releasing them as decomposed organic or already mineralized waste products. Macro-detritivores also modify the chemical and physical properties of soil, and redistribute materials between above- and belowground compartments [16]. These roles are expected to be substantial in deserts, because macro-detritivores are thought to be the main primary consumers in these systems [17,18]. Moreover, many desert macro-detritivores avoid the extreme surface conditions by burrowing belowground for extended time periods, where they deliver large quantities of fragmented litter, egesta and excretions [19]. More stable temperatures, higher moisture and the excreted nutrients' availability belowground provide favourable conditions for decomposers [20].

Since the 1970s, ecologists have echoed Noy-Meir's hypothesis [6,21], but empirical attempts to verify it were scant and came mostly from work on termites [7,22]. In the Chihuahuan desert, termites are responsible for 50% of the leaf litter removal [23–25] and root litter mass loss is fourfold higher in soils with termites than in soils without them [26]. Interestingly, other studies from this same system revealed no significant effects of termites on leaf litter mass loss [27,28]. In the Namib desert, termites account for 65% of buried detritus loss [29]. Other studies found that detritivores increase plant litter mass loss by a factor of 1.23 in the hot and dry Baza basin [30], but have no effect in a semi-arid Patagonian steppe [31]. To the best of our knowledge, other than these attempts, Noy-Meir's long-standing hypothesis has not received the attention it deserves.

Our main goal was to evaluate the role that macro-detritivores play in regulating plant litter cycling, focusing on the abundant desert isopod *Hemilepistus reaumuri*. Using litter baskets varying in hole sizes to regulate the entry of micro-, meso- and macrofauna, we demonstrated that macro-detritivores govern litter clearing in our desert field site. Moreover, we show that the isopods' belowground activity led to high nutrient concentrations around their burrows. Our results suggest that macro-detritivores are key regulators of litter decomposition and nutrient cycling in deserts.

## 2. Material and methods

### (a) Research system

The study was conducted in the Even-Ari Research Station, Central Negev Desert, Israel. This ecosystem is characterized by warm dry summers (May–September) and cool winters (November–February) when most of the precipitation occurs. The mean annual precipitation is 92 mm, and it highly fluctuates between consecutive years. The annual precipitation during the field experiment (August 2016 to August 2017) was 37.8 mm, less than one-third of the preceding year's 119.8 mm. Daily temperatures fluctuate tremendously between day and night (electronic supplementary material, appendix S1). The site is dominated by isolated patches of the perennial shrub *Haloxylon scoparium*, surrounded by biological soil crust (BSC). This shrub sheds many of its leaves from May to July. Macro-detritivores are very abundant in this site, and include termites (Isoptera), darkling beetles (Tenebrionidae), snails and the desert isopod *H. reaumuri*.

*Hemilepistus reaumuri* is a very abundant macro-detritivore in the deserts of the Middle East and North Africa. Isopod density in the Negev Desert can reach up to 48 individuals per 1 $m^2$ but fluctuates substantially between years, probably in response to soil moisture profile changes. Isopods are monogamous and live with their approximately 70 offspring in a single permanent

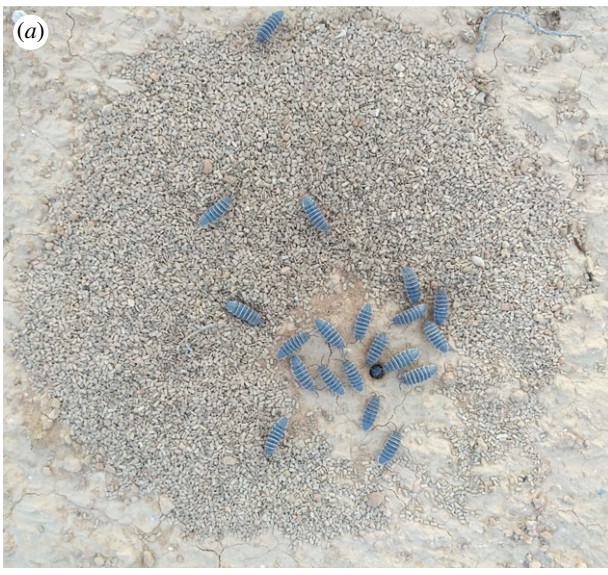

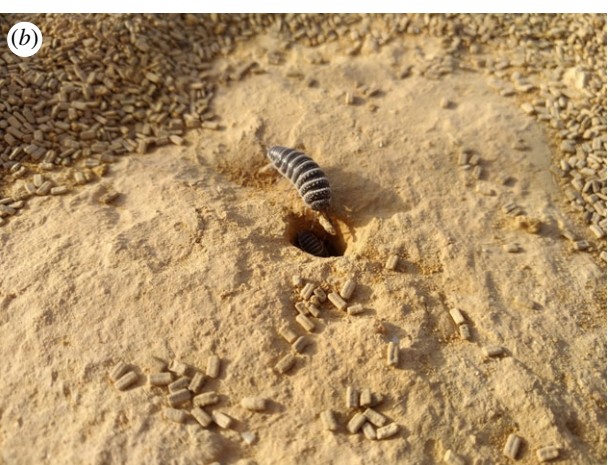

**Figure 1.** (*a*) A typical desert isopod burrow with a faecal pellet mound that surrounds the burrow entrance. The photo was taken just after dawn, when isopods evacuate faeces from the burrow. (*b*) A desert isopod bringing a piece of *Haloxylon scoparium* litter into the burrow.

burrow. Isopods remain belowground from November to February. They then leave their natal burrows, form pairs and establish new family burrows [32,33]. Foraging activity peaks in the hot and dry summer months, where isopods forage aboveground for plant litter and BSC during a short time window of 1–2 h after dawn [34]. During this time, isopods feed aboveground. Only at the end of their daily foraging do isopods often carry a small piece of plant litter into their burrows. Given that an individual isopod ingests approximately 4.74 mg leaf litter per day [35], and assuming 90% of the individuals actively forage in a given August morning [32], the consumption of litter by a typical *H. reaumuri* family can reach 300 mg d$^{-1}$. About 70% of the ingested litter is egested as faecal pellets [35], mostly within the burrow. Every morning before resuming foraging, isopods evacuate the faecal pellets and pile them in a circular mound around the burrow entrance (figure 1*a*). Towards the end of the activity season, faeces evacuation ceases despite ongoing feeding activity. Consequently, large quantities of faecal pellets remain within the burrows. Terrestrial isopods excrete gaseous ammonia as their main waste product, mostly during their belowground resting period [36]. In preliminary measurements, we found that an individual desert isopod excretes $NH_3$ at an average rate of 39 µg N day$^{-1}$. We also found that concentrations of gaseous ammonia directly above isopod burrows were similar to ambient concentrations, suggesting that most of the excreted ammonia remains belowground.

## (b) The contribution of macro-detritivores to surface litter removal

To quantify the relative role that macro-detritivores play in clearing plant litter, we placed litter baskets with different hole sizes at four consecutive time periods that correspond with variations in climatic conditions and isopod phenology. In our site, senesced leaves of *H. scoparium* bushes constitute the main litter resource for isopods. We therefore located 30 similar size *H. scoparium* bushes. Around each bush, we placed three litter basket types that differed in mesh sizes, as follows: micro-baskets allowing entry of only microorganisms (less than 200 µm), meso-baskets allowing entry of microorganisms and mesofauna (less than 2 mm), and macro-baskets that were identical to the meso-baskets with additional four side openings ($1 \times 10$ cm) approximately 1 cm above the basket floor. These elevated windows force macro-detritivores to climb in order to enter but minimizes accidental litter spill (for detailed descriptions see electronic supplementary material, appendix S2). All three basket types were placed at a distance of 10 cm from the bush, with the directions randomly assigned. To prevent baskets movement or flip over, we tightly anchored the baskets to the soil using arched metal stakes. The baskets were placed for four consecutive trial periods of 82, 68, 77 and 75 days, starting on August 2016. These periods correspond with late summer, winter, spring and early summer. The baskets contained dry *H. scoparium* litter that was collected in our field site just before each trial. This small, tubular and rather heavy litter tends to roll on the BSC and accumulates in small depressions adjacent to the bush, and is unlikely to be lifted and blown by the wind (see electronic supplementary material, appendix S2). We manually sorted the litter to remove woody particles, stones and animal remains, and homogenized the litter. The amount of litter used differed according to the 'fresh' litter availability. We used $4 \text{ g} \pm 0.1$ mg in the 1st and 4th trial periods, and $1 \text{ g} \pm 0.1$ mg in the 2nd and 3rd trial periods. To determine the dry litter mass, we oven-dried subsamples at 60°C for 48 h and weighed them in a semi-micro balance (Mettler Toledo MS105DU). To prevent accidental litter loss during transportation, we inserted each litter basket into a new zipper storage bag (see electronic supplementary material, appendix S2 for detailed protocol). At the end of each period, the baskets were collected and transported to the laboratory. We then oven dried the litter at 60°C for 48 h and weighed the samples. Seven (six in late summer and one in winter) of the 360 baskets used throughout the experiment were excluded from analysis, due to potential litter spill while handling. The litter removal rate was calculated as the difference between the initial and final dry litter mass, divided by the length of the trial in days. We did not correct our litter removal estimations for ash content. Thus, our estimations are likely to be somewhat less than the actual rates due to possible dust deposition, but the effect is probably similar across litter basket types due to their similar design.

## (c) Soil characteristics around isopod burrows in the field

To understand how isopod litter consumption propagates and affects soil properties, we sampled the environment surrounding 15 isopod burrows both above- and belowground. Samples were collected in May 2016, two to three months after the isopods dispersed from the burrows. To sample the faeces mound encircling each burrow, we gently removed and collected the mound using a scraper. As a no-isopod-faeces control, we gently removed and collected 2 mm thick BSC at a distance of 5 cm from the soil mound's outer circumference. We then removed the remainder BSC at a 30 cm radius from the burrow and collected 30 cm deep soil cores at distances of 0, 10 and 20 cm from the vertical burrow (electronic supplementary material, appendix S2). Samples were sieved (1 mm) to remove stones and plant material. Then, the pH, electrical conductivity (EC; a proxy for salinity), gravimetric moisture content, microbial biomass and concentrations of available $PO_4$, $NO_3$–N and $NH_4$–N were measured. The pH and EC were measured in a $1:1$ soil–water ratio solution, using electrodes (Mettler Toledo S220 and S230 for pH and EC, respectively). Microbial biomass was measured using the substrate-induced respiration (SIR) method [37]. Soil samples were extracted in 0.5 M $NaHCO_3$ for $PO_4$ and in 2 M KCl for inorganic N [38]. Nutrient concentrations were determined using colorimetric methods, following Murphy & Riley [39] for available $PO_4$, Kempers & Zweers [40] for $NH_4$–N and Norman & Stucki [41] for $NO_3$–N.

## (d) Nitrogen distribution around isopod burrows—laboratory microcosms

To reveal how isopod activity control nutrient distribution belowground, we constructed five custom-made Plexiglas microcosms that allow detailed monitoring of isopod activity and nutrient distribution belowground (hereafter 'isopolis'; for technical details see electronic supplementary material, appendix S2). Each isopolis was made of an aboveground chamber (dimensions $60 \times 30 \times 20$ cm) connected to a narrow ($60 \times 40 \times 0.8$ cm) belowground transparent chamber. We filled the aboveground chamber with 5 cm deep homogenized field-collected soil and paved it with field-collected BSC. The belowground chamber was compacted with the same homogenized soil, to which we translocated one *H. reaumuri* mating pair. Only two females produced offspring (approx. 30 individuals), therefore we replaced each of the non-productive pairs with 25–35 siblings taken from the field during early July. Every week, fresh *H. scoparium* litter was collected from the field and distributed on the BSC. The transparent belowground chambers were covered by opaque screens. Twice a week throughout the experiment (from 13 July to 29 September) we temporarily removed the opaque screens to record the distribution of isopods and faeces within the burrows. We used presence/absence data to determine the isopods and faeces frequencies of occurrence at each 5 cm depth point, to the burrow maximal depth. Two isopolises were excluded from this analysis due to soil remains that obstructed our view. At the end of the experiment, we carefully removed the front wall of the belowground chamber and sampled the soil, at 1 cm intervals from the burrow wall, to a distance of 10 cm. We repeated this sampling protocol from the surface to the burrow maximal depth in 5 cm intervals (see figure 4; electronic supplementary material, table S2-1; appendix S2). The amount of soil collected did not allow to determine inorganic nutrient concentrations. Therefore, the samples were lyophilized and grounded, and the total N content was determined using Micro-Dumas combustion analysis [42].

## (e) Statistical analysis

We used a linear mixed model (LMM) to evaluate whether rates of litter removal differ between the three basket types across all four trial periods, while accounting for spatial heterogeneity in the organismal activity. We ran an LMM with the litter removal rate as the response variable, basket type and trial period as fixed effects, and by-bush random intercept and by-bush random slope for basket type. To test for significance of the fixed effects, we used likelihood ratio tests (LRTs), comparing our model to LMMs not containing each of the fixed factors. A similar analysis was performed with initial litter mass as a random factor, for which the results were qualitatively the same (see electronic supplementary material, appendix S3). We then used a series of LMMs to analyse whether the rates of litter removal differ between the three basket types in each trial, separately. For each trial, we ran an LMM with the basket type as the fixed

**Table 1.** Results of statistical analyses for the litter baskets experiment. (*a*) Likelihood ratio tests (LRTs) testing the effects of litter basket type, trial and their interaction on litter removal rates. (*b*) LRTs testing the effect of litter basket type on litter removal rates in each trial separately. (*c*) Pairwise comparisons between litter removal rates in different litter basket types.

(*a*) *LRTs for all data*[a]

| effect | dAIC | $\chi^2$ | d.f. | *p*-value |
| --- | --- | --- | --- | --- |
| litter basket type | 128.3 | 144.3 | 8 | <0.0001 |
| trial | 103.9 | 121.8 | 9 | <0.0001 |
| interaction | 92.1 | 104.6 | 6 | <0.0001 |

(*b*) *LRTs for each trial separately*[b]

| trial | dAIC | $\chi^2$ | *n* | *p*-value |
| --- | --- | --- | --- | --- |
| late summer | 93.11 | 97.64 | 84 | <0.0001 |
| winter | 53.91 | 57.92 | 89 | <0.0001 |
| spring | 67.22 | 71.21 | 90 | <0.0001 |
| early summer | 9.99 | 13.99 | 90 | <0.001 |

(*c*) *Post hoc comparisons (Tukey)*

| trial | basket types compared | difference (mg d$^{-1}$) | d.f. | *t* | *p*-value |
| --- | --- | --- | --- | --- | --- |
| late summer | macro−micro | 8.87 ± 0.78 | 81 | 11.38 | <0.0001 |
| | macro−meso | 9.32 ± 0.8 | 81 | 11.62 | <0.0001 |
| | meso−micro | −0.45 ± 0.8 | 81 | −0.56 | 0.84 |
| winter | macro−micro | 1.06 ± 0.11 | 57.11 | 9.35 | <0.0001 |
| | macro−meso | 0.32 ± 0.11 | 57.6 | 2.78 | <0.05 |
| | meso−micro | 0.74 ± 0.11 | 57.6 | 6.48 | <0.0001 |
| spring | macro−micro | 2.31 ± 0.26 | 87 | 8.96 | <0.0001 |
| | macro−meso | 2.26 ± 0.26 | 87 | 8.77 | <0.0001 |
| | meso−micro | 0.05 ± 0.26 | 87 | 0.19 | 0.98 |
| early summer | macro−micro | 3.69 ± 1.16 | 87 | 3.19 | <0.01 |
| | macro−meso | 3.95 ± 1.16 | 87 | 3.42 | <0.01 |
| | meso−micro | −0.27 ± 1.16 | 87 | −0.23 | 0.97 |

[a]$n = 353$.
[b]d.f. = 2 in all trials.

effect and a by-bush random intercept. Inclusion of by-bush random slopes in these models was not applicable due to over-parameterization, but running these analyses using a different algorithm with by-bush random slopes yielded similar results (see electronic supplementary material, appendix S3). We tested the significance of the fixed effect using LRTs. *Post hoc* comparisons were done using Tukey's *p*-value adjustment.

We used LMMs to test whether soil properties differ as a function of the distance from the burrow, while accounting for variability among burrows in baseline values. For each soil property, we ran an LMM with the soil property as the response variable, the distance from the burrow as a fixed effect, and with by-burrow random intercepts. The significance of the distance effect was tested using LRTs. *Post hoc* comparisons were done using Tukey's *p*-value adjustment. We tested the differences between mound and BSC properties using paired Student's *t*-tests. Data of moisture and EC in mounds and soil crust and of moisture and $NO_3^-$ content in soil were log-transformed to meet the assumptions of paired *t*-tests and LMM, respectively.

We ran an LMM to test whether soil total N content in laboratory microcosms is affected by distance from the burrow, and by isopods and faeces frequencies of occurrence. The model included distance, isopod frequency and faeces frequency as fixed effects, as well as by-isopolis random intercept. N content data were log-transformed to meet the assumptions

of LMM. The significance of the fixed effects was tested using LRTs.

All tests were applied using $\alpha = 0.05$. LRTs and *post hoc* tests were applied using the Satterthwaite's d.f. approximation. For further details concerning statistical analysis, see electronic supplementary material, appendix S3.

## 3. Results

### (a) The contribution of macro-detritivores to surface litter removal

The trial period, the litter basket type and the interaction between them affected the litter removal rate (table 1*a*). Exploring the basket type effect in each trial separately revealed significant differences in all trials (table 1*b* and figure 2). *Post hoc* comparisons show that during all seasons, the rate of litter removal from macro-baskets was significantly higher than from meso- and micro-baskets (table 1*c* and figure 2). No significant differences in litter removal rates were found between meso- and micro-baskets, except during the winter, where the removal rates from meso-baskets were higher than from micro-baskets (table 1*c* and figure 2).

**Table 2.** Results of statistical analyses for soil properties near *Hemilepistus reaumuri* burrows. (a) Paired *t*-tests between the properties of faeces mounds and adjacent soil crust. (b) Likelihood ratio tests (LRTs) testing the effect of distance from a burrow on soil properties. (c) Pairwise comparisons between soil properties at discrete distances from the burrow.

| property | (a) *Mounds* versus *crust* (paired *t-test*)[a] | | (b) *Distance from burrow* (LRTs)[b] | | |
| --- | --- | --- | --- | --- | --- |
| | *t* | *p*-value | dAIC | $\chi^2$ | *p*-value |
| moisture | 3.6 | <0.01 | −1.06 | 2.94 | 0.23 |
| salinity (electric conductivity) | 5.58 | <0.0001 | −0.09 | 3.91 | 0.14 |
| pH | −1.65 | 0.12 | −2.94 | 1.05 | 0.59 |
| microbial biomass (SIR) | 3.41 | <0.01 | 1.31 | 5.31 | 0.07 |
| $NO_3$–N | 6.27 | <0.0001 | 288.88 | 9.22 | <0.01 |
| $NH_4$–N | 4.48 | <0.001 | 2.07 | 6.07 | <0.05 |
| available $PO_4$ | 0.38 | 0.71 | 6.93 | 10.92 | <0.01 |

| (c) *Post hoc comparisons (Tukey)* | | | | | |
| --- | --- | --- | --- | --- | --- |
| property | distances compared (cm) | difference (µg g$^{-1}$ soil) | d.f. | *t* | *p*-value |
| $NO_3$–N | 0–20 | $0.89 \pm 0.29$[c] | 28 | 3.12 | <0.05 |
| | 0–10 | $0.3 \pm 0.29$[c] | 28 | 1.06 | 0.54 |
| | 10–20 | $0.59 \pm 0.29$[c] | 28 | 2.06 | 0.12 |
| $NH_4$–N | 0–20 | $8.21 \pm 3.35$ | 42 | 2.45 | <0.05 |
| | 0–10 | $3.41 \pm 3.35$ | 42 | 1.02 | 0.57 |
| | 10–20 | $4.8 \pm 3.35$ | 42 | 1.43 | 0.33 |
| available $PO_4$ | 0–20 | $1.48 \pm 0.42$ | 27.14 | 3.51 | <0.01 |
| | 0–10 | $0.76 \pm 0.43$ | 27.33 | 1.75 | 0.2 |
| | 10–20 | $0.72 \pm 0.43$ | 27.33 | 1.67 | 0.23 |

[a]d.f. = 14, *n* = 30 for all tests.
[b]d.f. = 2, *n* = 45 for all tests, except available $PO_4$ (*n* = 44).
[c]Log-transformed data.

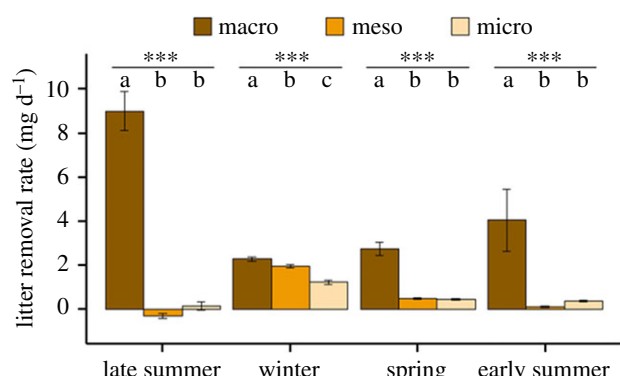

**Figure 2.** Comparison of plant litter removal rates (mean ± s.e.) from the micro-, meso- and macro-litter baskets during different periods of experimental exposure (trial periods). Asterisks represent the significance level from LRT tests (*p* < 0.001). Letters represent significant differences between groups within each trial period.

### (b) Soil characteristics around *Hemilepistus reaumuri* natural burrows

Values of moisture, EC, microbial biomass (as assessed by SIR) and concentrations of $NO_3$–N and $NH_4$–N were significantly higher in the faecal pellet mounds than in the BSC outside of the mounds, while pH and available $PO_4$ did not differ between treatments (table 2*a* and figure 3*a*).

The belowground concentrations of $NO_3$–N, $NH_4$–N and available $PO_4$ decreased with increasing distance from isopod burrows (table 2*b* and figure 3*b*). However, the microbial biomass, pH, moisture and EC were not significantly affected by the distance from the burrows (table 2*b* and figure 3*b*). *Post hoc* comparisons indicate that all measured soil nutrient concentrations were significantly higher near the burrows than at a distance of 20 cm from the burrows (table 2*c* and figure 3*b*). However, nutrient concentrations did not significantly differ between soils near the burrows and at 10 cm from the burrows, nor between soils at 10 and 20 cm distances from the burrows (table 2*c* and figure 3*b*).

### (c) Association between isopod activity and N distribution

Our laboratory experiment showed elevated N concentrations in the homogenized soil surrounding the isopod burrows. At intermediate burrow depth, the elevated N concentrations were detectable even at a horizontal distance of up to 10 cm from the burrow walls, but above and below this depth the effect wore off at a shorter distance (figure 4*a*). Isopod frequency of occurrence in the three analysed isopolises displayed a unimodal distribution with a maximum at medium depths, whereas faeces frequency of occurrence increased with depth, peaking at the bottom of the burrow (figure 4*b*). LRTs yielded significant effects of distance from burrow and isopod frequency of occurrence on soil total N

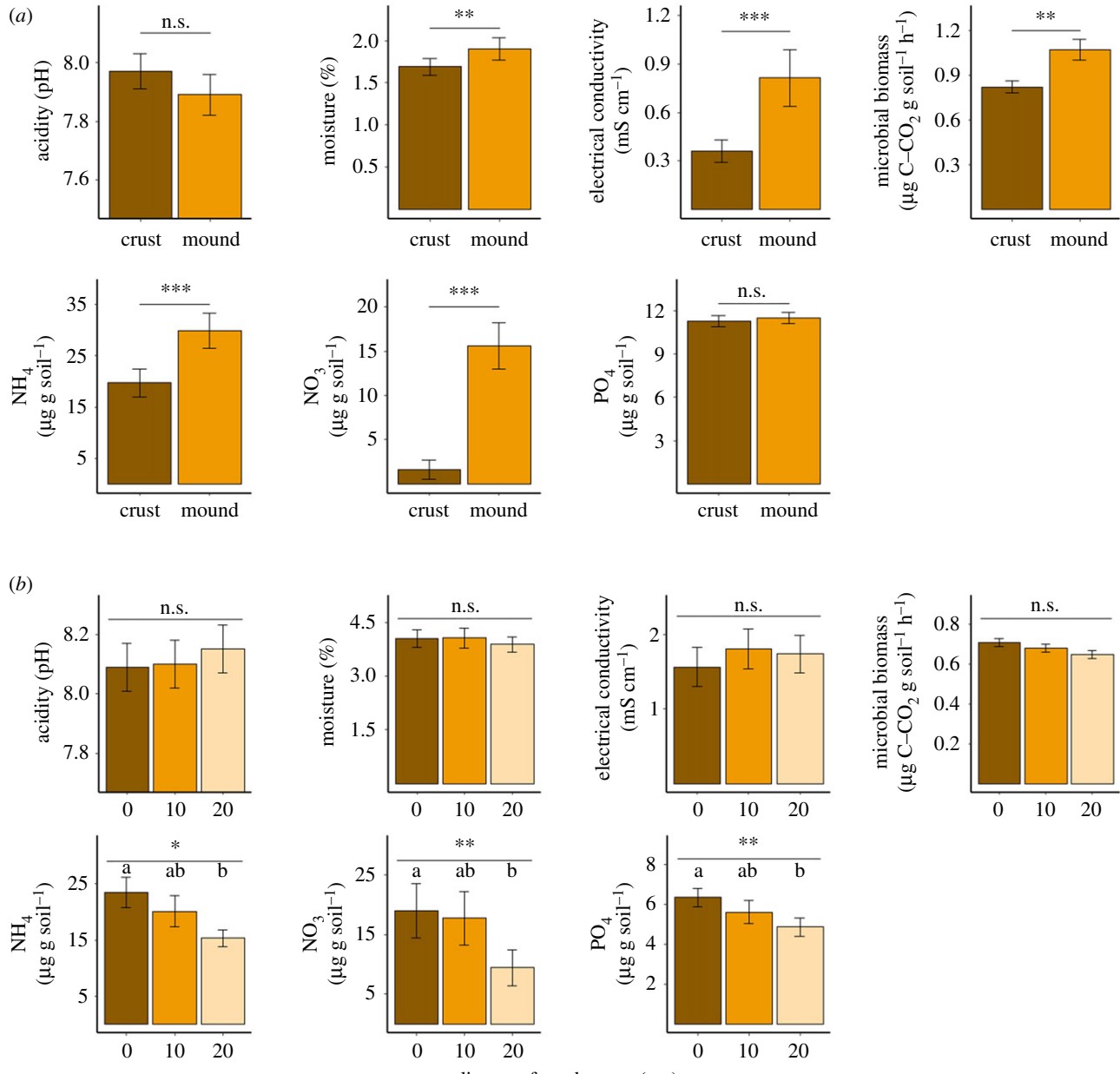

**Figure 3.** Comparison of soil properties (mean ± s.e.) (a) between the isopod faecal pellet mounds and the surrounding BSC and (b) at the upper 30 cm of the soil profile between 0, 10 and 20 cm distances from the burrow entrance. Asterisks represent the significance level from LRT tests (*$p < 0.05$, **$p < 0.01$, ***$p < 0.001$; n.s., non-significant). Letters represent significant differences between groups.

content. Faeces frequency of occurrence did not significantly affect soil N content (table 3).

## 4. Discussion

We have shown that macro-detritivores, especially desert isopods, are key nutrient cycling regulators in our Negev Desert field site. Using litter baskets with different mesh sizes, we found that macro-detritivores dominate litter clearing from the desert surface. This effect was most evident during the dry summer when isopods are most active, and to a much lesser extent during the winter when isopods are inactive and the aboveground microbial and mesofaunal activity peaks. We also found substantial enrichment of inorganic nutrients around isopod burrows both at the soil surface and in subsoils, suggesting that isopods enhance mineralization of the harvested plant litter nutrients.

Empirical evidence for the dryland decomposition conundrum [7] comes mostly from litter-bag experiments [4]. This

is possibly why attempts to resolve this conundrum focused on mechanisms that affect *in situ* abiotic and biotic decomposition within the realm of microbial litter bags (e.g. [8,11]). The role of bigger detritivores did not get enough attention despite convincing arguments that macro-decomposition may be pivotal for understanding nutrient cycling in arid lands [1,17].

Macro-detritivores are probably the main primary consumers in many deserts for two reasons [17]. First, desert systems often have limited herbivory due to the spatial and temporal scarcity of green plant material [17]. Second, macro-detritivores have physiological and behavioural adaptations that allow them to remain active and forage even during the hot, dry desert summers [19]. Indeed, over the four trial periods, macro-detritivores in our field site accounted for 89% of the plant litter clearing. Rates of litter clearing by macro-detritivores were closely associated with the phenology of desert isopods, peaking in the summer when isopods are most active and declining in the winter

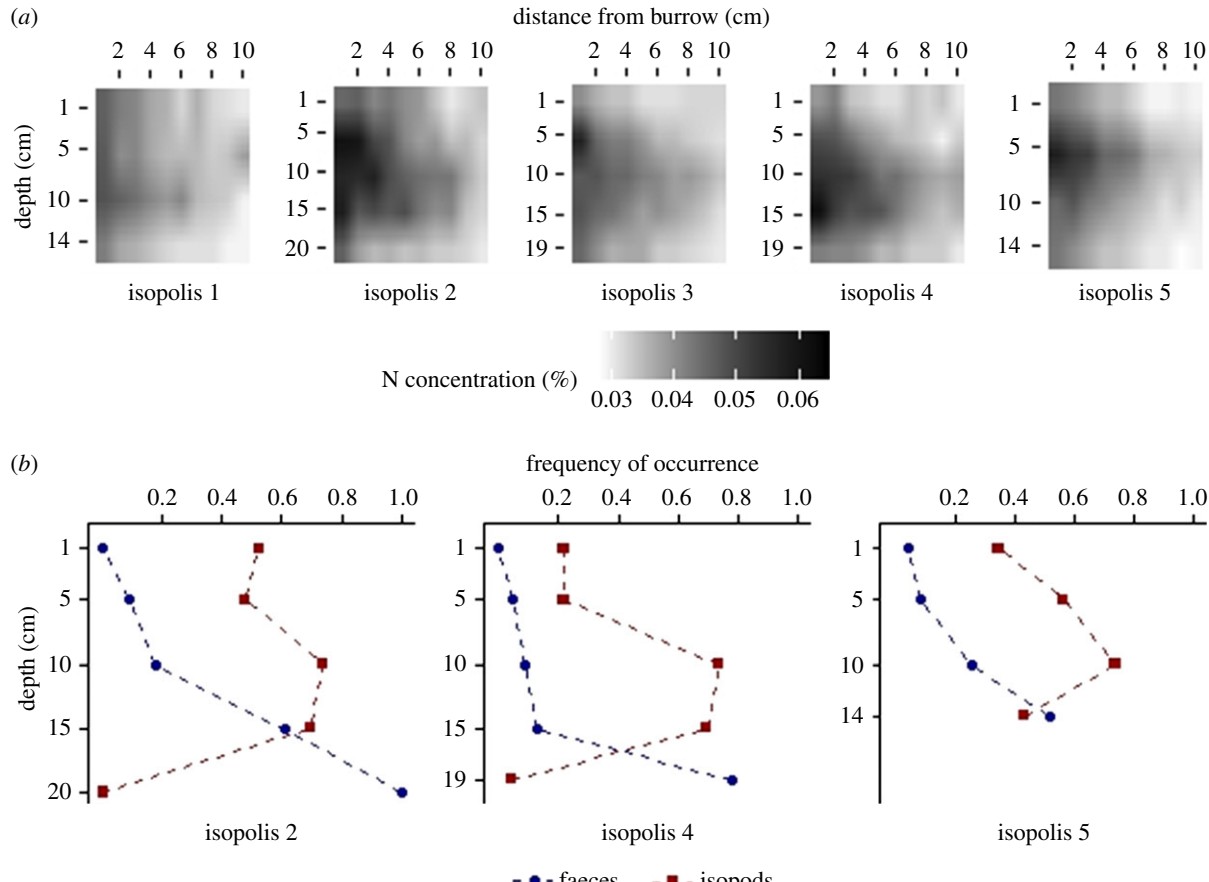

**Figure 4.** Soil profiles of (*a*) linearly interpolated N concentrations and (*b*) relative occurrence of isopods (red squares) and faeces (blue circles) in the lab microcosms (isopolises).

**Table 3.** Results of likelihood ratio tests for the effects of distance from the burrow, isopod frequency of occurrence and faeces frequency of occurrence on soil total N content in microcosms experiment ($n = 140$).

| effect | dAIC | $\chi^2$ | d.f. | *p*-value |
|---|---|---|---|---|
| distance from burrow | 187.079 | 189.08 | 1 | <0.0001 |
| isopod frequency of occurrence | 77 | 81.937 | 1 | <0.0001 |
| faeces frequency of occurrence | 1.54 | 0.4671 | 1 | 0.49 |

when isopods stay belowground. Average rates of litter removal in early summer 2017 were much lower than in late summer 2016, possibly due to a sharp decline in the local isopod population, attributed to a severe drought in the 2016–2017 rainy season. However, the low litter clearing rates observed in summer 2017 were still substantially higher than the rates achieved by micro- and mesofauna combined, even during the winter when their activity peaks.

It is important to note that our macro-baskets were designed to balance the need to minimize litter losses by wind, and the need to guarantee free entry to macro-detritivores. Thus, we cannot rule out the possibility that our quantitative results are slightly affected by unnaturally low macro-arthropod activity or by accidental litter losses. We took every precaution to minimize accidental litter spill (see methods) and found absolutely no evidence for litter losses by wind (see electronic supplementary material, appendix S1). Thus, we expect this possible deviation from the natural clearing rates to be minimal. Similarly, our litter basket design may somewhat alter the complex inter-actions between abiotic weathering agents, litter moisture,

soil–litter mixing and decomposers. For instance, the mesh surrounding the basket may partially block solar radiation and wind from reaching the litter. This common drawback of the litter-bag approach may slightly affect our quantitative estimations, but is unlikely to affect them qualitatively, due to the high similarity in basket design between the three treatments, and because of the findings' large effect size.

Macro-detritivores play a crucial role in plant litter clear-ing. An extensive meta-analysis on decomposition in terrestrial ecosystems suggests that macrofauna significantly increased plant litter loss, with a mean effect size (Hedges's D) of approximately 0.25 [15]. However, this meta-analysis was limited to mostly temperate ecosystems, and may under-estimate the effects of macrofauna in desert systems. Our results revealed that macro-arthropods in the Negev Desert have a much greater role in clearing plant litter than the global average calculated by Frouz *et al.* [15], with mean Hedges's D [43] values of 2.55, 0.70, 1.83 and 0.71 for late summer, winter, spring and early summer, respectively. Yet our results fall well within the effect size range found in studies from other deserts focusing on termites [23–25,29].

Another meta-analysis that explored the effect of soil fauna on litter decomposition, without differentiating between organism size classes, included a few studies from arid ecosystems [44]. This comprehensive synthesis found that excluding fauna reduced the rate of litter decomposition by 35% globally, but only by 18% when focusing on cold and dry systems that were pooled together. García-Palacios et al. [44] explained this difference by suggesting that biological activity in these systems is constrained by temperature and/or moisture. In our study, while micro- and mesofauna were indeed constrained by moisture, the activity of macro-detritivores peaked during the hot and dry season at the utmost availability of plant litter. These results suggest that the role that macro-detritivores play in regulating litter cycling is independent of aboveground soil–litter moisture [45]. If these results apply to other drylands, then macro-detritivores may be the key biological agent of decomposition in hot deserts. Their role is hypothesized to be less important in cold deserts because of temperature constraints that may limit macro-detritivore abundance [31,46].

Isopod activity was found to regulate plant litter mineralization in our field site, generating hotspots of mineralized nutrients both at the surface and the sub-soil surrounding their burrows. Soil, faecal pellets and uneaten detritus are evacuated from the isopod burrows and mounded around them. These loose mounds were more saline and slightly moister than the surrounding soil crust. Soil microbes prospered in the mounds that were much richer in nutrients, with 1.5-fold increase in ammonium and 9.8-fold increase in nitrate, compared with adjacent soil crust. We detected elevated levels of inorganic nutrient concentrations around isopod burrows also below the surface. Isopod activity resulted in 1.5-fold increase in ammonium, twofold increase in nitrate and 1.3-fold increase in phosphate concentrations near the vertical burrow compared with 20 cm away from the burrow. Interestingly, nutrient enrichment was not limited to the burrow walls, but extended up to a distance of about 10 cm from the vertical burrows. Our laboratory results supported these findings, showing elevated N levels up to 10 cm from the isopod burrow walls. The field burrows were 15 months old at the time of the sampling, after the isopods had already abandoned them. Thus, the elevated concentrations of inorganic nutrients around isopod burrows, both at the surface and in the sub-soil, imply hastened mineralization processes.

Desert isopods transport large quantities of nutrients from the surface to their burrows in the forms of faecal matter and to a lesser extent fragmented detritus. Fragmented detritus and egested materials are expected to decompose faster than intact plant litter [20,47] (but see [48]). High moisture and stable temperatures within the burrows provide favourable conditions for decomposer activity that may further promote plant litter decomposition [20]. Thus, we suggest that fragmentation and digestion of plant nutrients, and the transportation of these nutrients belowground, contributed to the accelerated rates of litter mineralization near the isopods' burrows.

Isopods can also propel nutrient cycling by ingesting and assimilating plant litter nutrients, and by excreting these nutrients already in mineralized forms like carbon dioxide and gaseous ammonia [16,36]. Ammonia gas has a tremendous affinity for water [49]. Thus, the excreted ammonia may be adsorbed instantaneously in the moist burrow walls and then oxidized to nitrate by nitrifying microorganisms. Our laboratory measurements suggest that this pathway probably governs the mineralization of plant litter nutrients, especially until the burrow abandonment. We found that the highest N enrichment was detected in mid-burrow depth and not just below the surface, as was expected if N leaching from the faecal pellet mound was governing the distribution of plant-litter N. We also found that the distribution of N in the soil profile was not associated with the accumulation of faecal pellets and organic debris near the burrow floor. Nitrogen distribution in the soil was positively associated with the location of isopods within the burrow. To the best of our knowledge, this is the first example in which ingestion, assimilation and excretion of plant litter nutrients govern recycling rates of plant litter nutrients by macro-detritivores.

## 5. Conclusion

Our study supports Noy-Meir's longstanding hypothesis that macro-detritivores are important regulators of plant litter cycling in deserts [1]. Using litter baskets with different mesh sizes, we have shown that macro-detritivores are pivotal in clearing plant detritus from the desert surface. This role, which was independent of ground surface moisture, exceeded the combined effect of microorganisms, mesofauna and abiotic agents of degradation, and was larger than the relative contribution of macro-detritivores to litter clearing in more mesic systems. We have also found elevated concentrations of inorganic nutrients in the vicinity of the desert isopod burrows. This finding implies that isopods accelerate the decomposition of the harvested litter by releasing assimilated plant nutrients already in inorganic forms, and by processing litter nutrients and transporting them belowground, where the conditions are more favourable for decomposers. Thus, isopod burrows serve as conduits of nitrogen and phosphorus into the soil that may become islands of fertility in an otherwise poor environment. While clearly demonstrating the potential of macro-detritivores to drive nutrient cycling in deserts, further research is required before macro-detritivores can be included in global models of litter decomposition. This should include extensive assessments of the generality and overall importance of our findings by conducting similar experiments in other arid lands. In addition, the duration of such experiments should be extended, to capture temporal variations in litter conditions and macro-detritivore abundance. To gain better mechanistic understandings, stable-isotope pulse-chase approaches can be used to meticulously track the different pathways by which macro-detritivores affect litter cycling in deserts. This field research should be complimented by detailed laboratory and field experimentation aimed at testing and quantifying the individual pathways.

Data accessibility. Data available from the Dryad Digital Repository: https://doi.org/10.5061/dryad.6nq98f5 [50].

Authors' contributions. D.H. and N.S. conceived this study and designed the experiments. N.S. collected and analysed the data. All authors participated in writing the manuscript, contributed critically to the drafts and gave final approval for publication.

Competing interests. We declare we have no competing interests.

Funding. This article was supported by European Research Council (grant no. ERC-2013-StG-337023 (ECOSTRESS)) and by the Israel Science Foundation (grant no. ISF-1471/12) to D.H.

Acknowledgements. We thank Mark Bradford for his most valuable advice, three anonymous reviewers who offered helpful suggestions for improving the manuscript, Rita Dumbur for laboratory assistance and Moshe Zaguri for preliminary data.

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
