## [Reviewer comments · Proceedings of the Royal Society B: Biological Sciences]

Review History

RSPB-2019-1118.R0 (Original submission)

Review form: Reviewer 1

Recommendation

Major revision is needed (please make suggestions in comments)

Scientific importance: Is the manuscript an original and important contribution to its field?

Good

General interest: Is the paper of sufficient general interest?

Good

Quality of the paper: Is the overall quality of the paper suitable?

Acceptable

Is the length of the paper justified?

Yes

Should the paper be seen by a specialist statistical reviewer?

Yes

Do you have any concerns about statistical analyses in this paper? If so, please specify them explicitly in your report.

No

It is a condition of publication that authors make their supporting data, code and materials available - either as supplementary material or hosted in an external repository. Please rate, if applicable, the supporting data on the following criteria.

Is it accessible?

Yes

Is it clear?

Yes

Is it adequate?

Yes

Do you have any ethical concerns with this paper?

No

Comments to the Author

Sagi et al. experimentally examined the effect of macro-detritivore (isopods) on plant litter removal and soil nutrient concentrations in a desert ecosystem. They found that the isopods contributed most to the litter removal and improved the soil nutrient level. This manuscript is well written, and the high rate (>80%) of litter removal by the isopods impressed me very much.

I have two major concerns. One is that the macro-detritivores are primarily responsible for the plant litter removal but not necessarily for the nutrient release from the litter. Although the isopods released some nutrients into soil after digestion, fecal pellets should contain most of the nutrients; but the fecal pellets are transported to surface (Figure 1) and likely take long time for total decomposition. In this end, the title and the conclusion about the macro-detritivores controlling the desert cycling is NOT accurate. Perhaps something like 'macro-detritivore facilitate litter removal and improve soil nutrient' is more accurate albeit not attractive. Otherwise, the authors need to calculate how much nutrients had directly released from isopods to soil.

The other is that it could be highly possible for the isopods to lose litters during their transportation from litter bags to their burrows. I am wondering whether how much of the removed litter was really transported to the burrows. The experimental design did not consider this factor and the discussion did not either. In addition, the discussion section

In general I like this topic, but I think the authors might have overestimated the importance of macro-detritivores (NOT comparable to termites in litter decomposition) to the nutrient cycling of this particular ecosystem.

Review form: Reviewer 2

Recommendation

Major revision is needed (please make suggestions in comments)

Scientific importance: Is the manuscript an original and important contribution to its field?

Good

General interest: Is the paper of sufficient general interest?

Good

Quality of the paper: Is the overall quality of the paper suitable?

Acceptable

Is the length of the paper justified?

Yes

Should the paper be seen by a specialist statistical reviewer?

No

Do you have any concerns about statistical analyses in this paper? If so, please specify them explicitly in your report.

No

It is a condition of publication that authors make their supporting data, code and materials available - either as supplementary material or hosted in an external repository. Please rate, if applicable, the supporting data on the following criteria.

Is it accessible?

Yes

Is it clear?

Yes

Is it adequate?

No

Do you have any ethical concerns with this paper?

No

Comments to the Author

To study the promotion of burrowing detritivores to litter removal and nutrient cycling in desert ecosystem, this research created a field decomposition experiment and a laboratory microcosm. The equipment in this research is excellent, three litter baskets and Plexiglas microcosms are powerful to study litter removal and isopod behavior.

But the relevance of field experiment and laboratory microcosm are not close; research indexes are less; mechanism study is absent; results are lack of comprehensive. The promoted effect of burrowing isopod to nutrient cycling can't be explained exactly by results and discussion in this research.

There still are some questions about your research.

a) How did the rainfall influence the burrows of *Hemilepistus reaumuri* during the raining?

Although the rainfall was very low.

b) When did the litter fall of *Haloxylon scoparium* reach peaks? Were the litter fall parallel to the detritivores' activity, especially in summer?

c) Your results reported that the highest nitrogen content detected in mid-burrows? What is the mechanism?

d) The paper mentioned that many macro-detritivores live in patches of *H. scoparium*, what's the

population status of these detritivores? What's the effect of them on litter removal? What was the exact contribution of *H. reaumuri* compared with other detritivores?

Besides, this paper will be better if the following questions corrected.

1) Mistaken or imprecise expression.

P3 Line 25-26 "Attempts to reveal...alternative moisture sources." It isn't an appropriate expression;

P3 Line 29 "Quantified" is not so appropriate;

P4 Line 42-43 There are too many key words, and key words are not so precise. (up to 6 key words);

P6 Line 91 An "of" should be added behind "role";

P10 Line 178 "Soils", it's better to replace "soils" with "soil samples";

P12 Line 226 "NO₃ content", imprecise expression;

P12 Line 237 The title cannot generalize content of that paragraph exactly, "Litter removal of different basket treatments" or analogous may be better;

P15 Line 293 2016-17, imprecise expression;

P16 Line 326 An "of" should be added behind "role";

P32 Figure 3 The format of graph are unequal, see the bottom border of pH ;

2) Language needs modification, some sentences are too long to understand, it will be better if these sentences were segmented.

P9 Line 172-175 The long sentence should be divided to avoid logical misunderstand.

P12 Line 228-229 The long sentence should be divided to avoid logical misunderstand.

P14 Line 270-273 The long sentence should be divided to avoid logical misunderstand.

P15 Line 283-295 The language of this part is not native;

P15 Line 308-311 The language of this part is not native;

3) References are outdated. The references from 2014-2019 are less than 25%, and there are too many references which were published before 2000. Besides, format of references needs to be uniform.

P7 Line 114-130 references of this part are too old, fresh data should be considered;

P10 Line 179 Format of references are wrong;

P16 Line 330 Format of references are wrong;

P17 Line 349 Format of references are wrong;

4) Discussion is lack of comprehensive, and arrangement of references in discussion is unreasonable. Some contents are lack of references, it makes part of views in discussion are not so objective.

P16 Line 332-346 References are absent;

P17 Line 357 References are absent;

P18 Line 365 This sentence will be more precise if some limited terms added; (see Macro-detritivore identity drives leaf litter diversity effects, OIKOS, 2011; Do woodlice and earthworms interact synergistically in leaf litter decomposition? FUNCTIONAL ECOLOGY, 2005)

5) Some questions in Materials and Method.

P8 Line 142 "With the directions (east, west and south)", why chose these three directions?

P9 Line 175 When pH was determined, why the ratio of soil samples to water is 1:1 were chosen? As we known, the ratio of 1:2.5 or 1:3 is a more common choice.

P11 Line 203 References or simply description is absent.

Finally, your research will be more wonderful if the nutritional flow of "litter - isopod - burrows - soil" be study exactly.

To study the promotion of burrowing detritivores to litter removal and nutrient cycling in desert ecosystem, the authors conducted a field experiment and a laboratory microcosm. The equipment design is good. Three litter baskets and Plexiglas microcosms are powerful to study litter removal and isopod behavior.

However, the results are lack of comprehensive. The promoted effect of burrowing isopod to nutrient cycling can't be explained exactly by results and discussion in this research. And the author should answer following questions

- a) How did the rainfall influence the burrows of *Hemilepistus reaumuri* during the raining? Although the rainfall was very low.
- b) When did the litter fall of *Haloxylon scoparium* reach peaks? Were the litter fall parallel to the detritivores' activity, especially in summer?
- c) Your results reported that the highest nitrogen content detected in mid-burrows? What is the mechanism?
- d) The paper mentioned that many macro-detritivores live in patches of *H. scoparium*, what's the population abundance of these detritivores? What's the effect of them on litter removal? What was the exact contribution of *H. reaumuri* compared with other detritivores?

Some minor mistakes are as follows:

- 1) Mistaken or imprecise expression.

P4 Line 42-43 There are too many key words, and key words are not so precise. (up to 6 key words);

P10 Line 178 "Soils", it's better to replace "soils" with "soil samples";

P15 Line 293 2016-17, imprecise expression;

P32 Figure 3 The format of graph are unequal, see the bottom border of pH ;

- 2) Format of references needs to be uniform.

P10 Line 179 Format of references are wrong;

P16 Line 330 Format of references are wrong;

P17 Line 349 Format of references are wrong;

- 3) Discussion is lack of comprehensive. Some contents are lack of references, it makes part of views in discussion are not so objective.

P16 Line 332-346 References are absent ;

P17 Line 357 References are absent ;

P18 Line 365 References are absent ;

- 4) Some questions in Materials and Method.

P8 Line 142 "With the directions (east, west and south)", why chose these three directions?

P9 Line 175 When pH was determined, why the ratio of soil samples to water is 1:1 were chosen? As we known, the ratio of 1:2.5 or 1:3 is a more common choice.

P11 Line 203 References or simply description is absent.

Decision letter (RSPB-2019-1118.R0)

04-Jul-2019

Dear Mr Sagi:

I am writing to inform you that your manuscript RSPB-2019-1118 entitled "Burrowing detritivores propel nutrient cycling in a desert ecosystem" has, in its current form, been rejected for publication in Proceedings B.

This action has been taken on the advice of two referees, who have recommended that substantial revisions are necessary. One has provided very detailed comments and another mainly notes that macro-detritivores need to be better addressed in some way. With this in mind we would be happy to consider a resubmission, provided the comments of the referees are fully addressed. However please note that this is not a provisional acceptance.

In your revision process, please take a second look at how open your science is; our policy is that all data involved with the study should be made openly accessible-- see: <https://royalsociety.org/journals/ethics-policies/data-sharing-mining/>
Insufficient sharing of data can delay or even cause rejection of a paper.

Sincerely,

Professor John Hutchinson, Editor
mailto: proceedingsb@royalsociety.org

Reviewer(s)' Comments to Author:

Referee: 1

Comments to the Author(s)

Sagi et al. experimentally examined the effect of macro-detritivore (isopods) on plant litter removal and soil nutrient concentrations in a desert ecosystem. They found that the isopods contributed most to the litter removal and improved the soil nutrient level. This manuscript is well written, and the high rate (>80%) of litter removal by the isopods impressed me very much.

I have two major concerns. One is that the macro-detritivores are primarily responsible for the plant litter removal but not necessarily for the nutrient release from the litter. Although the isopods released some nutrients into soil after digestion, fecal pellets should contain most of the nutrients; but the fecal pellets are transported to surface (Figure 1) and likely take long time for

total decomposition. In this end, the title and the conclusion about the macro-detritivores controlling the desert cycling is NOT accurate. Perhaps something like 'macro-detritivore facilitate litter removal and improve soil nutrient' is more accurate albeit not attractive. Otherwise, the authors need to calculate how much nutrients had directly released from isopods to soil.

The other is that it could be highly possible for the isopods to lose litters during their transportation from litter bags to their burrows. I am wondering whether how much of the removed litter was really transported to the burrows. The experimental design did not consider this factor and the discussion did not either. In addition, the discussion section

In general I like this topic, but I think the authors might have overestimated the importance of macro-detritivores (NOT comparable to termites in litter decomposition) to the nutrient cycling of this particular ecosystem.

Referee: 2

Comments to the Author(s)

To study the promotion of burrowing detritivores to litter removal and nutrient cycling in desert ecosystem, this research created a field decomposition experiment and a laboratory microcosm. The equipment in this research is excellent, three litter baskets and Plexiglas microcosms are powerful to study litter removal and isopod behavior.

But the relevance of field experiment and laboratory microcosm are not close; research indexes are less; mechanism study is absent; results are lack of comprehensive. The promoted effect of burrowing isopod to nutrient cycling can't be explained exactly by results and discussion in this research.

There still are some questions about your research.

- a) How did the rainfall influence the burrows of *Hemilepistus reaumuri* during the raining? Although the rainfall was very low.
 - b) When did the litter fall of *Haloxylon scoparium* reach peaks? Were the litter fall parallel to the detritivores' activity, especially in summer?
 - c) Your results reported that the highest nitrogen content detected in mid-burrows? What is the mechanism?
 - d) The paper mentioned that many macro-detritivores live in patches of *H. scoparium*, what's the population status of these detritivores? What's the effect of them on litter removal? What was the exact contribution of *H. reaumuri* compared with other detritivores?
- Besides, this paper will be better if the following questions corrected.

1) Mistaken or imprecise expression.

P3 Line 25-26 "Attempts to reveal...alternative moisture sources." It isn't an appropriate expression;

P3 Line 29 "Quantified" is not so appropriate;

P4 Line 42-43 There are too many key words, and key words are not so precise. (up to 6 key words);

P6 Line 91 An "of" should be added behind "role";

P10 Line 178 "Soils", it's better to replace "soils" with "soil samples";

P12 Line 226 "NO₃ content", imprecise expression;

P12 Line 237 The title cannot generalize content of that paragraph exactly, "Litter removal of different basket treatments" or analogous may be better;

P15 Line 293 2016-17, imprecise expression;

P16 Line 326 An "of" should be added behind "role";

P32 Figure 3 The format of graph are unequal, see the bottom border of pH

;

2) Language needs modification, some sentences are too long to understand, it will be better if these sentences were segmented.

P9 Line 172-175 The long sentence should be divided to avoid logical misunderstand.

P12 Line 228-229 The long sentence should be divided to avoid logical misunderstand.

P14 Line 270-273 The long sentence should be divided to avoid logical misunderstand.

P15 Line 283-295 The language of this part is not native;

P15 Line 308-311 The language of this part is not native;

3) References are outdated. The references from 2014-2019 are less than 25%, and there are too many references which were published before 2000. Besides, format of references needs to be uniform.

P7 Line 114-130 references of this part are too old, fresh data should be considered;

P10 Line 179 Format of references are wrong;

P16 Line 330 Format of references are wrong;

P17 Line 349 Format of references are wrong;

4) Discussion is lack of comprehensive, and arrangement of references in discussion is unreasonable. Some contents are lack of references, it makes part of views in discussion are not so objective.

P16 Line 332-346 References are absent;

P17 Line 357 References are absent;

P18 Line 365 This sentence will be more precise if some limited terms added; (see Macro-detritivore identity drives leaf litter diversity effects, OIKOS, 2011; Do woodlice and earthworms interact synergistically in leaf litter decomposition? FUNCTIONAL ECOLOGY, 2005)

5) Some questions in Materials and Method.

P8 Line 142 "With the directions (east, west and south)", why chose these three directions?

P9 Line 175 When pH was determined, why the ratio of soil samples to water is 1:1 were chosen? As we known, the ratio of 1:2.5 or 1:3 is a more common choice.

P11 Line 203 References or simply description is absent.

Finally, your research will be more wonderful if the nutritional flow of "litter - isopod - burrows - soil" be study exactly.

To study the promotion of burrowing detritivores to litter removal and nutrient cycling in desert ecosystem, the authors conducted a field experiment and a laboratory microcosm. The equipment design is good. Three litter baskets and Plexiglas microcosms are powerful to study litter removal and isopod behavior.

However, the results are lack of comprehensive. The promoted effect of burrowing isopod to nutrient cycling can't be explained exactly by results and discussion in this research. And the author should answer following questions

a) How did the rainfall influence the burrows of *Hemilepistus reaumuri* during the raining? Although the rainfall was very low.

b) When did the litter fall of *Haloxylon scoparium* reach peaks? Were the litter fall parallel to the detritivores' activity, especially in summer?

c) Your results reported that the highest nitrogen content detected in mid-burrows? What is the mechanism?

d) The paper mentioned that many macro-detritivores live in patches of *H. scoparium*, what's the population abundance of these detritivores? What's the effect of them on litter removal? What was the exact contribution of *H. reaumuri* compared with other detritivores?

Some minor mistakes are as follows:

1) Mistaken or imprecise expression.

P4	Line	42-43	There are too many key words, and key words are not so precise. (up to 6 key words);
P10	Line	178	“Soils”, it’s better to replace “soils” with “soil samples”;
P15	Line	293	2016-17, imprecise expression;
P32	Figure	3	The format of graph are unequal, see the bottom border of pH ;
2)			Format of references needs to be uniform.
P10	Line	179	Format of references are wrong;
P16	Line	330	Format of references are wrong;
P17	Line	349	Format of references are wrong;
3)			Discussion is lack of comprehensive. Some contents are lack of references, it makes part of views in discussion are not so objective.
P16	Line	332-346	References are absent ;
P17	Line	357	References are absent ;
P18	Line	365	References are absent ;
4)			Some questions in Materials and Method.
P8	Line	142	“With the directions (east, west and south)”, why chose these three directions?
P9	Line	175	When pH was determined, why the ratio of soil samples to water is 1:1 were chosen? As we known, the ratio of 1:2.5 or 1:3 is a more common choice.
P11	Line	203	References or simply description is absent.

Author's Response to Decision Letter for (RSPB-2019-1118.R0)

See Appendix A.

RSPB-2019-1647.R0

Review form: Reviewer 2

Recommendation

Accept with minor revision (please list in comments)

Scientific importance: Is the manuscript an original and important contribution to its field?

Acceptable

General interest: Is the paper of sufficient general interest?

Acceptable

Quality of the paper: Is the overall quality of the paper suitable?

Good

Is the length of the paper justified?

Yes

Should the paper be seen by a specialist statistical reviewer?

Yes

Do you have any concerns about statistical analyses in this paper? If so, please specify them explicitly in your report.

Yes

It is a condition of publication that authors make their supporting data, code and materials available - either as supplementary material or hosted in an external repository. Please rate, if applicable, the supporting data on the following criteria.

Is it accessible?

Yes

Is it clear?

Yes

Is it adequate?

Yes

Do you have any ethical concerns with this paper?

No

Comments to the Author

The highest nitrogen happened in the mid-burrows, the mechanism is still unclear, needing further discussing it.

Decision letter (RSPB-2019-1647.R0)

12-Aug-2019

Dear Mr Sagi:

Your manuscript has now been peer reviewed and the reviews have been assessed by an Associate Editor. The reviewers' comments (not including confidential comments to the Editor) and the comments from the Associate Editor are included at the end of this email for your reference. As you will see, the reviewer and the Editor have raised some concerns with your manuscript and we would like to invite you to revise your manuscript to address them. The point made is brief, but important, so it should be dealt with rigorously.

Research ethics:

Use of animals and field studies:

All supplementary materials accompanying an accepted article will be treated as in their final form. They will be published alongside the paper on the journal website and posted on the online figshare repository. Files on figshare will be made available approximately one week before the

accompanying article so that the supplementary material can be attributed a unique DOI. Please try to submit all supplementary material as a single file.

Please submit a copy of your revised paper within three weeks. If we do not hear from you within this time your manuscript will be rejected. If you are unable to meet this deadline please let us know as soon as possible, as we may be able to grant a short extension.

Best wishes,
Professor John Hutchinson, Editor
mailto:proceedingsb@royalsociety.org

Reviewer(s)' Comments to Author:

Referee: 2

Comments to the Author(s).

The highest nitrogen happened in the mid-burrows, the mechanism is still unclear, needing further discussing it.

Author's Response to Decision Letter for (RSPB-2019-1647.R0)

See Appendix B.

RSPB-2019-1647.R1 (Revision)

Review form: Reviewer 1 (Shucun Sun)

Recommendation

Accept as is

Scientific importance: Is the manuscript an original and important contribution to its field?

Acceptable

General interest: Is the paper of sufficient general interest?

Acceptable

Quality of the paper: Is the overall quality of the paper suitable?

Acceptable

Is the length of the paper justified?

Yes

Should the paper be seen by a specialist statistical reviewer?

Yes

Do you have any concerns about statistical analyses in this paper? If so, please specify them explicitly in your report.

Yes

It is a condition of publication that authors make their supporting data, code and materials available - either as supplementary material or hosted in an external repository. Please rate, if applicable, the supporting data on the following criteria.

Is it accessible?

Yes

Is it clear?

Yes

Is it adequate?

Yes

Do you have any ethical concerns with this paper?

No

Comments to the Author

I appreciate the changes the authors made and I think this is a clearly-presented paper indicating the importance of macro-detritivores to desert nutrient cycling.

I am afraid that Meso (<2m) should be Meso (<2mm). Please double check.

Review form: Reviewer 3

Recommendation

Accept as is

Scientific importance: Is the manuscript an original and important contribution to its field?

Excellent

General interest: Is the paper of sufficient general interest?

Excellent

Quality of the paper: Is the overall quality of the paper suitable?

Excellent

Is the length of the paper justified?

Yes

Should the paper be seen by a specialist statistical reviewer?

No

Do you have any concerns about statistical analyses in this paper? If so, please specify them explicitly in your report.

No

It is a condition of publication that authors make their supporting data, code and materials available - either as supplementary material or hosted in an external repository. Please rate, if applicable, the supporting data on the following criteria.

Is it accessible?

No

Is it clear?

N/A

Is it adequate?

N/A

Do you have any ethical concerns with this paper?

No

Comments to the Author

I reviewed this manuscript for another journal. I enjoyed reading it again and think the authors have done a nice job revising it to address the concerns I raised in the first version. I don't have any further comments except for one small wording comment below.

----- Line Comments -----

Line 324: What is "plant-litter mineralization?" Do you mean mineralized N originating from plant litter or the mineralization of N within decomposing plant litter?

Decision letter (RSPB-2019-1647.R1)

04-Oct-2019

Dear Mr Sagi

I am pleased to inform you that your manuscript RSPB-2019-1647.R1 entitled "Burrowing detritivores regulate nutrient cycling in a desert ecosystem" has been accepted for publication in Proceedings B. Congratulations!

The referee(s) have recommended publication, but also suggest some very minor revisions to your manuscript. Therefore, I invite you to respond to the referee(s)' comments and revise your manuscript. Because the schedule for publication is very tight, it is a condition of publication that you submit the revised version of your manuscript within 7 days. If you do not think you will be able to meet this date please let us know. It is just a couple of words needing fixing.

[http://datadryad.org/submit?journalID=RSPB&manu=\(Document not available\)](http://datadryad.org/submit?journalID=RSPB&manu=(Document+not+available)) which will take you to your unique entry in the Dryad repository. If you have already submitted your data to dryad you can make any necessary revisions to your dataset by following the above link. Please see <https://royalsociety.org/journals/ethics-policies/data-sharing-mining/> for more details.

Sincerely,

Professor John Hutchinson
Editor, Proceedings B
<mailto:proceedingsb@royalsociety.org>

Reviewer(s)' Comments to Author:

Referee: 1

Comments to the Author(s)

I appreciate the changes the authors made and I think this is a clearly-presented paper indicating the importance of macro-detritivores to desert nutrient cycling.

I am afraid that Meso (<2m) should be Meso (<2mm). Please double check.

Referee: 3

Comments to the Author(s)

I reviewed this manuscript for another journal. I enjoyed reading it again and think the authors have done a nice job revising it to address the concerns I raised in the first version. I don't have any further comments except for one small wording comment below.

----- Line Comments -----

Line 324: What is "plant-litter mineralization?" Do you mean mineralized N originating from plant litter or the mineralization of N within decomposing plant litter?

Author's Response to Decision Letter for (RSPB-2019-1647.R1)

See Appendix C.

Decision letter (RSPB-2019-1647.R2)

07-Oct-2019

Dear Mr Sagi

I am pleased to inform you that your manuscript entitled "Burrowing detritivores regulate nutrient cycling in a desert ecosystem" has been accepted for publication in Proceedings B.

Open Access

You are invited to opt for Open Access, making your freely available to all as soon as it is ready for publication under a CCBY licence. Our article processing charge for Open Access is £1700. Corresponding authors from member institutions (<http://royalsocietypublishing.org/site/librarians/allmembers.xhtml>) receive a 25% discount to these charges. For more information please visit <http://royalsocietypublishing.org/open-access>.

Paper charges

Sincerely,

Editor, Proceedings B
<mailto:proceedingsb@royalsociety.org>

Appendix A

Dear Professor Hutchinson,

Thank you for the opportunity to revise our manuscript RSPB-2019-1118 entitled "Burrowing detritivores propel nutrient cycling in a desert ecosystem" for further consideration by Proceedings of the Royal Society B. We found the comments and suggestion for improvement made by two anonymous reviewers very valuable, and we have done our best to address them. We have inserted our responses alongside with the original comments (our responses are preceded by ** and in blue font).

Referee: 1

Sagi et al. experimentally examined the effect of macro-detritivore (isopods) on plant litter removal and soil nutrient concentrations in a desert ecosystem. They found that the isopods contributed most to the litter removal and improved the soil nutrient level. This manuscript is well written, and the high rate (>80%) of litter removal by the isopods impressed me very much.

** We appreciate the reviewer clear and very supportive summary. Yet, we must emphasize that we measured the overall litter removal by all macro-detritivores combined and cannot exclude the role that termite or beetles play in addition to isopods. Throughout the text we were very meticulous not to infer that isopods were the only regulators of litter clearing by using the term "macro-detritivores" and not "isopods" wherever the terms "litter clearing" or "litter removal" was used. For example, in lines 263-266 we wrote: "Indeed, over the four trial periods, macro-detritivores in our field site accounted for 89% of plant-litter clearing. Rates of litter clearing by macro-detritivores were tightly associated with the phenology of desert isopods, peaking in the summer when isopods are most active and declining in the winter when isopods stay belowground." Following the reviewer comment we re-checked the entire text and can verify that we always use "macro-detritivores" and not isopods to explain the litter mass loss from the litter baskets.

I have two major concerns. One is that the macro-detritivores are primarily responsible for the plant litter removal but not necessarily for the nutrient release from the litter. Although the isopods released some nutrients into soil after digestion, fecal pellets should contain most of the nutrients; but the fecal pellets are transported to surface (Figure 1) and likely take long time for total decomposition. In this end, the title and the conclusion about the macro-detritivores controlling the desert cycling is NOT accurate. Perhaps something like 'macro-detritivore facilitate litter removal and improve soil nutrient' is more accurate albeit not attractive. Otherwise, the authors need to calculate how much nutrients had directly released from isopods to soil.

** We agree with the reviewer that the first sentence of the conclusion was too decisive. Consequently, we toned it down to more accurately reflect our findings (lines 341-342). In addition, we changed our title to "Burrowing detritivores regulate nutrient cycling in a desert ecosystem", so that it doesn't make an overly strong statement about the rate of nutrient cycling. By altering the spatial location of the litter

and by changing its physical structure and molecular form, macro-arthropods regulate litter nutrient cycling in our system. Macro-arthropods cleared the litter from the litter-baskets much faster than all other mechanisms combined. Large proportion of it was ingested in the basket and only relatively small percentage was removed as fragmented litter to the burrow (we now better explain this important information in lines 320-321). After one season inorganic N in the fecal-pellets mound was 114% higher than in the surrounding soil crust implying there is a significant nutrient release from the feces. Moreover, inorganic N in burrow walls was 71% higher than in soil 20 cm away from the burrow, and in available Phosphate we detected a similar pattern of 30% increase. Our lab experiment also contributes to this notion showing that in about 6 months we detected elevated levels of inorganic nitrogen up to a distance of 10 cm from the burrow walls and that this pattern was most likely due to excretion of nitrogen waste in the form of ammonia (lines 328-339).

The other is that it could be highly possible for the isopods to lose litters during their transportation from litter bags to their burrows. I am wondering whether how much of the removed litter was really transported to the burrows. The experimental design did not consider this factor and the discussion did not either. In addition, the discussion section

**** We can safely determine that litter loss by isopods on their way back to the burrow is negligible. Isopods transport fragmented litter only on their way back to the burrow at the end of the daily activity time. Isopods' spend about 1-2 hr a day feeding above ground and retreat to the burrow often with one small piece of litter. Isopods do not shuttle between the food patch and the burrows like ants do. Thus, isopods carry only very small portion of the consumed litter. Moreover, watching thousands of isopods retreating to the burrow we can safely conclude that almost all the small litter fragments that are carried by isopods are getting safely to the burrow. We now better explain this point in lines 93-96.**

In general I like this topic, but I think the authors might have overestimated the importance of macro-detrivores (NOT comparable to termites in litter decomposition) to the nutrient cycling of this particular ecosystem.

**** We agree with this general comment and toned down our statements to more accurately reflect our findings (e.g., title, line 305, lines 341-342). In our system isopods seems to be the most important regulators of litter clearing – see calculations in lines 97-99). Yet, as aforementioned, and is clearly stated in the title we focus on macro-detrivores that include termites and other species and do not infer that isopods were the only species responsible for the very high rate of litter clearing. We demonstrated the fate of these nutrients using the very abundant macro-detrivores in our system the desert isopods (~48 individuals per m²).**

Referee: 2

Comments to the Author(s)

To study the promotion of burrowing detritivores to litter removal and nutrient cycling in desert ecosystem, this research created a field decomposition experiment and a laboratory microcosm. The equipment in this research is excellent, three litter baskets and Plexiglas microcosms are powerful to study litter removal and isopod behavior.

But the relevance of field experiment and laboratory microcosm are not close; research indexes are less; mechanism study is absent; results are lack of comprehensive. The promoted effect of burrowing isopod to nutrient cycling can't be explained exactly by results and discussion in this research.

** We apologize but cannot fully understand this comment. We clearly showed that macro-detritivores are responsible for 89% of the litter clearing from the desert surface. Using the isopods to demonstrate the fate of litter removal we showed elevated nutrient concentration around their burrows. This means that in less than a year large proportion of those nutrients were already mineralized. In a complement mechanistic lab experiment we showed that isopods propel litter mineralization most likely by assimilating the litter nutrients and excreting them as gaseous ammonia. To better explain this point, we added calculations regarding the ammonia excretion per isopods per day (lines 105-106). We also toned down the title and our conclusions in order to accurately reflect our findings (line 305, lines 341-342).

There still are some questions about your research.

- a) How did the rainfall influence the burrows of *Hemilepistus reaumuri* during the raining?
Although the rainfall was very low.

** As far as we know very little rain is penetrating the burrow during run-off events. This is mostly because many burrows are surrounded by the fecal pellet mounds that divert the run-off from the burrow entrance. There is a clear association between isopod burrow survival and precipitation amounts in the prior season.

- b) When did the litter fall of *Haloxylon scoparium* reach peaks? Were the litter fall parallel to the detritivores' activity, especially in summer?

** The main *Haloxylon scoparium* new growth happens during February- March. The bushes begin shedding leaves mostly from the end of May to mid- July. We now better emphasize this important point in the methods (Lines 85-86).

- c) Your results reported that the highest nitrogen content detected in mid-burrows? What is the mechanism?

** We elaborate on this point in lines 339-350 and Figure 4. The intermediate depth is where most isopods stay while in the burrows. Fecal-pellets and litter debris accumulate at the bottom of the burrow (see Fig 4b). Thus, one could expect to see the highest nutrient concentration there. The fact that the highest N concentration was in the intermediate depth suggest that ammonia excretion govern this pattern. Leaching from the fecal pellet mound would generate a vertical gradient, hence is unlikely. This mechanistic component of the MS is important because it emphasizes that ingestion and digestion

of the litter play a key role in accelerating plant litter decomposition in this system. We explain these points in lines 328-339.

- d) The paper mentioned that many macro-detritivores live in patches of *H. scoparium*, what's the population status of these detritivores? What's the effect of them on litter removal? What was the exact contribution of *H. reaumuri* compared with other detritivores?

** We changed the wording to better explain the macro-detritivores inhabit the site and not specifically the patch (lines 86-87). We did not monitor population dynamics of any of the macro-detritivores and could not quantify the distinct contribution of each species. This is why we use the term macro-detritivores and not isopods when discussing litter clearing from the litter-baskets.

Besides, this paper will be better if the following questions corrected.

- 1) Mistaken or imprecise expression.

P3 Line 25-26 "Attempts to reveal...alternative moisture sources." It isn't an appropriate expression;

** We do not understand why?

P3 Line 29 "Quantified" is not so appropriate;

** We do not agree with this comment. We meticulously measured the mass loss in each litter basket type. Thus, we quantified the relative contribution of each organism size class to litter loss.

P4 Line 42-43 There are too many key words, and key words are not so precise. (up to 6 key words);

** We apologize for this mistake and reduce the key-words number to six.

P6 Line 91 An "of" should be added behind "role";

** Thank you, we changed to " the role that..."

P10 Line 178 "Soils", it's better to replace "soils" with "soil samples";

** Changed as suggested

P12 Line 226 "NO₃ content", imprecise expression;

** Changed to **NO₃⁻**

P12 Line 237 The title cannot generalize content of that paragraph exactly, "Litter removal of different basket treatments" or analogous may be better;

** We do not agree. The title is very accurate since the difference between the meso and macro basket reflects the contribution of macro-detritivores to litter removal.

P15 Line 293 2016-17, imprecise expression;

** We changed to 2016-2017

P16 Line 326 An “of” should be added behind “role”;

** Thank you, we changed to the “role that ...”

P32 Figure 3 The format of graph are unequal, see the bottom border of pH ;

** We decided not to start from zero in order to emphasize the differences.

2) Language needs modification, some sentences are too long to understand, it will be better if these sentences were segmented.

P9 Line 172-175 The long sentence should be divided to avoid logical misunderstand.

** We fixed the problem as follows: “Samples were sieved (1 mm) to remove stones and plant material. Then, the pH, electrical conductivity (EC; a proxy for salinity), gravimetric moisture content, microbial biomass and concentrations of available PO₄, NO₃-N and NH₄-N were measured.”

P12 Line 228-229 The long sentence should be divided to avoid logical misunderstand.

** We added a comma to fix the problem

P14 Line 270-273 The long sentence should be divided to avoid logical misunderstand.

** We do not see a need to break this sentence

P15 Line 283-295 The language of this part is not native;

** Since we are not native English speakers, we sent the manuscript before submission to a review by a professional scientific language editor. However, some mistakes or unclear wording may have remained. Thus, we now re-edited the paragraph mentioned based on the advice of a native English-speaking colleague.

P15 Line 308-311 The language of this part is not native;

** Since we are not native English speakers, we sent the manuscript before submission to a review by a professional scientific English editor. However, some mistakes or unclear wording may have remained. Thus, we now re-edited the paragraph mentioned based on the advice of a native English-speaking colleague.

3) References are outdated. The references from 2014-2019 are less than 25%, and there are too many references which were published before 2000. Besides, format of references needs to be uniform.

** We strongly disagree with this statement. As a general practice, it is better to cite older work as a way to give proper credit to the people responsible for the idea or original findings. More recent papers

should also be cited only if they provide additional information that does not exist in the older papers or provide comparative information. Even if a good review paper exists, it is highly recommended also to give credit to the studies that conceived the idea or discover the phenomenon. That said, we will highly appreciate more focal recommendations in case that we unintentionally missed important papers that could put our work in a better context.

P7 Line 114-130 references of this part are too old, fresh data should be considered;

** We are well aware of the new most exciting work on desert isopods (e.g., work by Ayari and colleagues). We decided to cite the older literature because of the reasons mentioned above and even more so because our site is in the exact same area in which this older data was collected. That said, we added two relevant references (lines 88-95)

P10 Line 179 Format of references are wrong;

** Thank you for this important comment. We edited all references to match the journal's citing format.

P16 Line 330 Format of references are wrong;

** Thank you for this important comment. We edited all references to match the journal's citing format.

P17 Line 349 Format of references are wrong;

** Thank you for this important comment. We edited all references to match the journal's citing format.

4) Discussion is lack of comprehensive, and arrangement of references in discussion is unreasonable. Some contents are lack of references, it makes part of views in discussion are not so objective.

** We do not understand this very general comment. We will appreciate more focal criticism that may help us improve the paper.

P16 Line 332-346 References are absent;

** This must be a misunderstanding since the information provided is summarizing our own findings from this paper and not information collected from other papers.

P17 Line 357 References are absent;

** We hypothesize this part based on N-cycling. We are not aware of a relevant paper but will highly appreciate a focal recommendation.

P18 Line 365 This sentence will be more precise if some limited terms added; (see Macro-detrivore identity drives leaf litter diversity effects, OIKOS, 2011; Do woodlice and earthworms interact synergistically in leaf litter decomposition? FUNCTIONAL ECOLOGY, 2005)

** These two excellent papers demonstrate that detritivore diversity can enhance litter decomposition. Like many other papers they state that feeding can affect decomposition rates. Our results however

indicate that ingestion, assimilation and excretion of gaseous ammonia is key to explain the very high rates of litter mineralization. As such it provides first evidence for the importance of this rather trivial process.

5) Some questions in Materials and Method.

P8 Line 142 “With the directions (east, west and south)”, why chose these three directions?

** There is no specific reason why those directions were chosen. We just picked those directions to add consistency to the design. We rotated the litter baskets so there will be no direction effect on litter clearing. We deleted the directions in order to decrease confusion.

P9 Line 175 When pH was determined, why the ratio of soil samples to water is 1:1 were chosen? As we known, the ratio of 1:2.5 or 1:3 is a more common choice.

** We agree that 1:2.5 ratio is a common choice, but could not find a good reasoning for why. The variation in the soil-water ratio that people use for measuring pH is very large and different labs or subfields use different ratios all the way from 1:1 to 1:5. Gregorich, E. G., & Carter, M. R. (2007) in *Soil sampling and methods of analysis* recommend that : “When measuring soil pH in water, the main concern is that an increase in the amount of water added will cause an increase in pH; it is therefore important to keep the ratio constant and as low as possible. However, the supernatant solution must be sufficient to immerse the electrode properly without causing too much stress when inserting the tip of the electrode into the soil and to allow the porous pin on the electrode to remain in the solution above the soil”. You can also see the link: <https://blog.hannainst.com/soil-ph-testing> that recommends a 1:1 ratio.

P11 Line 203 References or simply description is absent.

** Thank you, reference was added

Finally, your research will be more wonderful if the nutritional flow of “litter – isopod – burrows – soil” be study exactly.

** We completely agree with this recommendation. In this study we put emphasis on litter clearing by macro-detritivores and the overall effect of a focal macro-detritivore, the desert isopods, on nutrient enrichment around the burrows. In an ongoing experiment we are using ¹⁵N labeled litter to measure the actual pathway by which isopods affect N –cycling. We believe that these two projects reveal different angles on how macro-detritivores regulate nutrient cycling that together may shed new light on our understanding of litter decomposition in deserts.

Appendix B

Dear Prof. Hutchinson,

Thank you for the opportunity to revise our manuscript RSPB-2019-1647 entitled "Burrowing detritivores regulate nutrient cycling in a desert ecosystem" for further consideration by Proceedings of the Royal Society B. We have done our best to address the comment made by the 2nd referee. Please find below our response (preceded by **** and in blue font**) and the revised manuscript with tracked changes.

Reviewer(s)' Comments to Author:

Referee: 2

Comments to the Author(s).

The highest nitrogen happened in the mid-burrows, the mechanism is still unclear, needing further discussing it.

****We conducted the lab ISOPOLIS experiment to improve our mechanistic understanding of the ways by which isopods control nutrient distribution (line 160). In this experiment we found that subterranean soil near the burrow mid-depth was enriched in N: "At intermediate burrow depth the elevated N concentrations were detectable even at a horizontal distance of up to 10 cm from the burrow walls, but above and below this depth the effect wore off at a shorter distance (Fig. 4A)" (lines 235-238).**

Three mechanisms can lead to the observed nutrient enrichment of the subterranean soil. First, decomposition of organic matter (in the form of isopod feces and litter residues) within the burrow can lead to N enriched soil. This mechanism is expected to result in N accumulation near the deeper part of the burrow, where the organic matter piled up (lines 240-241; Fig. 4B), hence is not supported by our results ("Feces frequency of occurrence did not significantly affect soil N content (Table 3)"; lines 242-243). We elaborated on this point in lines 338-340.

Second, decomposition of fecal-pellets in the aboveground mound can lead to nutrient leaching from the surface to the subterranean soil. As referee 2 pointed out, our results show that nitrogen concentrations were highest at mid-burrow depth (lines 236-238; Fig. 4A), allowing to reject this hypothesis too (lines 334-336).

Third, isopods assimilation and excretion of plant-litter nutrients can lead to N enriched soil. Desert isopods like other terrestrial isopods excrete nitrogen in the form of gaseous ammonia (lines 103-105). We found and reported (lines 105-106) that an individual desert isopod excretes NH₃ at an average rate of 39 µg N per day. Desert isopods live in large family groups of ~70 individual within a single permanent burrow (line 91). Thus, we expected to find elevated ammonia levels just above the burrow entrance. We found and reported that ammonia levels above the burrow entrance did not differ from ambient ammonia levels (lines 106-107), suggesting that the assimilated plant litter-N is being excreted as gaseous ammonia and remain within the isopod burrow. Gaseous ammonia is considerably lighter than air and will rise in dry

air. However, because of ammonias tremendous affinity for water, it may be adsorbed instantaneously in the moist burrow walls and then oxidized to nitrate by nitrifying microorganisms. In figure 4b we clearly showed that isopods remain in the burrow mid-depth and that the fecal-pellets and other detritus were placed mostly in the lower part of the burrow. Thus, there is a correlation between the isopods position within the burrow and the N-soil enrichment ("LRTs yielded significant effects of distance from burrow and Isopod frequency of occurrence on soil total N content." lines 241-242). Based on these findings, we suggested that during the 3 months experiment, ammonia excretion best explained the spatial distribution of N-enrichment in the soil surrounding the burrow. We discuss these insights in great details in the last paragraph of the discussion (328-348), and suggest a possible mechanism for the biological conversion of gaseous ammonia to nitrate.

To better explain the purpose of the experiment and the suggested mechanism as was suggested by the 2nd reviewer, we completely revised the text (328-348). We explained why the alternative hypotheses were rejected and highlight the importance of our findings. We are not aware of existing literature or alternative hypotheses that can better explain our findings.

Appendix C

Dear Prof. Hutchinson,

Thank you for accepting our manuscript RSPB-2019-1647.R1 entitled "Burrowing detritivores regulate nutrient cycling in a desert ecosystem" for publication in Proceedings of the Royal Society B. We address the comments made by the referees. Please find below our response (preceded by **** and in blue font**) and the revised manuscript with tracked changes.

Reviewer(s)' Comments to Author:

Referee: 1

Comments to the Author(s)

I appreciate the changes the authors made and I think this is a clearly-presented paper indicating the importance of macro-detritivores to desert nutrient cycling.

I am afraid that Meso (<2m) should be Meso (<2mm). Please double check.

**** Thank you for drawing our attention for this mistake. This comment refers to Figure S2-2 in Appendix S2. We corrected to "Meso (<2mm)" and uploaded the edited appendix.**

Referee: 3

Comments to the Author(s)

I reviewed this manuscript for another journal. I enjoyed reading it again and think the authors have done a nice job revising it to address the concerns I raised in the first version. I don't have any further comments except for one small wording comment below.

----- Line Comments -----

Line 324: What is "plant-litter mineralization?" Do you mean mineralized N originating from plant litter or the mineralization of N within decomposing plant litter?

**** Thank you for this comment. We mean mineralized nutrients originating from plant litter. We rephrased this sentence in the revised manuscript (line 324) .**